# The Role of Vitamin D in the Management of Major Depressive Disorder: A Systematic Review

**DOI:** 10.3390/ph18060792

**Published:** 2025-05-25

**Authors:** Andreea Roșian, Mihaela Zdrîncă, Luciana Dobjanschi, Laura Grațiela Vicaș, Mariana Eugenia Mureșan, Camelia Maria Dindelegan, Rita Ioana Platona, Eleonora Marian

**Affiliations:** 1Doctoral School of Biomedical Sciences, Faculty of Medicine and Pharmacy, University of Oradea, 410087 Oradea, Romania; rosian.andreea@student.uoradea.ro (A.R.); emarian@uoradea.ro (E.M.); 2Department of Preclinical Discipline, Faculty of Medicine and Pharmacy, University of Oradea, 10, 1 December Square, 410073 Oradea, Romania; mzdrinca@uoradea.ro (M.Z.); dobjanschil@uoradea.ro (L.D.); mmuresan@uoradea.ro (M.E.M.); 3Department of Pharmacy, Faculty of Medicine and Pharmacy, University of Oradea, 10, 1 December Square, 410073 Oradea, Romania; 4Faculty of Social Sciences, University of Oradea, 3, Universitatii Street, 410087 Oradea, Romania; dindkamy@yahoo.com; 5Psychiatry Department, County Clinical Emergency Hospital of Oradea, 65, Gheorghe Doja Street, 410169 Oradea, Romania; ritaioanaplatona@yahoo.com

**Keywords:** depression, major depressive disorder, vitamin D, 25-hydroxyvitamin D

## Abstract

**Background/Objective:** Depression is a widespread and complex disorder, constituting a major public health concern due to its significant impact on mental health. Because of the limitations of major depressive disorder (MDD) treatment, recent research on depression management has focused on identifying new therapeutic strategies. The effects of vitamin D on the brain, mediated through various mechanisms, suggest the potential implication of vitamin D in the pathophysiology of depression. In this systematic review, our objective was to evaluate the correlation between serum levels of 25-hydroxyvitamin D (25(OH)D) and depression based on evidence from cross-sectional and cohort studies. Furthermore, we also assessed the effect of vitamin D supplementation in relation to depressive symptoms, using data from randomised controlled trials (RCTs). **Methods:** To achieve the proposed objective, we have compiled a report that includes a selection of empirical evidence necessary to review the relationship between vitamin D and depression. In this regard, relevant articles were searched on platforms such as PubMed, MDPI, ResearchGate, Springer Link, Springer Open, and ScienceDirect. A total of 13,976 records, published between 2008 and 2024, were initially identified through database searches. After the study selection process, performed according to the PRISMA guidelines, 70 articles were included in the systematic review. **Results:** According to most cross-sectional and cohort studies, the results highlight an inverse relationship between serum 25(OH)D levels and the risk of depression, as well as the severity of depressive symptoms. An increase in serum 25(OH)D concentration is associated with an improvement in depression test scores, with vitamin D supplementation exerting a beneficial effect on both the incidence and the prognosis of depression. **Conclusions:** Based on current evidence which indicates the implications of vitamin D in the neurobiological mechanisms associated with depression, and the results obtained in most of the studies, which demonstrate an inverse relationship between serum 25(OH)D levels and the beneficial effect of vitamin D supplementation on depressive symptoms, vitamin D could represent an adjunctive therapy in the management of MDD. More rigorous studies, without methodological errors, are needed to correctly and definitively assess the impact of vitamin D in relation to depression.

## 1. Introduction

Mood disorders, also known as affective disorders, are characterised by an external manifestation of disposition, that is, emotion experienced at an internal level, and involve a spectrum with the two extremities of mania and depression [1]. Depression is one of the most prevalent psychiatric disorders [2], and can manifest itself through different symptoms, such as depressive mood, sleep and appetite disturbances, alterations in body weight, loss of interest and pleasure in some activities, decreased concentration ability and reduced energy levels, the presence of feelings of guilt and worthlessness, and suicidal thoughts [3].

MDD can occur at any age, but the risk is significantly higher in the second and third decade of life, being more frequent in women than in men. The onset of a depressive episode may be influenced by environmental, cultural, and socioeconomic factors, family history, and the presence of medical or psychiatric comorbidities, as well as genetic predisposition [4].

With a lifetime prevalence of 17%, depression affects 300 million people worldwide [5], and is estimated to become the leading cause of disability globally by 2030 [6].

Although the exact cause has not yet been fully elucidated, there are some theories that attempt to explain the pathophysiological mechanism of MDD [7]. One of the most widely accepted theories explaining the aetiology of depression is the monoaminergic theory [8]. Oxidative stress, hippocampal and frontal lobe dysfunction, neurotoxicity, and inflammatory and immunological processes are complemental mechanisms with potential impacts on the pathophysiology of depression. In recent years, interest has grown in understanding the role of genetic and epigenetic factors [9].

Pharmacological treatment of MDD involves the use of antidepressant medications, whose mechanism targets the monoaminergic system; more precisely, it involves the correction of serotonin, norepinephrine, and dopamine deficiency [10]. RCTs have highlighted the beneficial effects of psychotherapy and its association with medication-based antidepressant treatment in patients with moderate-to-severe symptoms. Additionally, in severe forms of depression, with psychotic symptoms, it is possible to include an antipsychotic medication or electroconvulsive therapy (ECT) in the treatment plan [4].

In recent years, it has been difficult to demonstrate the efficacy of antidepressant medication compared to placebos in clinical trials [1,9]. The response rate to first-line medication is roughly 50% of patients, while for second-line therapy, it decreases to approximately 30% [11].

Recently, research in the field of MDD management has focused on the identification of new therapeutic strategies [12]. Among the main reasons for this increasing interest are the low response rate to pharmacological treatment [11], the significant adverse effects of medication, and the debilitating impact of depressive disorders on mental health, exacerbated by economic instability and home quarantine, which were consequences of the SARS-CoV-2 pandemic [12].

Vitamin D is a fat-soluble vitamin [13], known for its role in calcium and phosphorus homeostasis and in the mineralisation process [14]. Interest in the extraskeletal effects of vitamin D has increased, such that numerous studies have focused on its potential to influence cell growth, immune system activity, and metabolic function [15]. Furthermore, scientific evidence suggests that it is also essential for the development and functioning of the brain [16].

The discovery of a correlation between total intracranial volume (TIV) and serum vitamin D concentration in MDD patients led researchers to conduct neuroimaging studies monitoring the potential effect of vitamin D, or its deficiency, on the brain. Hypovitaminosis D has been associated with a smaller volume of the brain and larger lateral cerebral ventricles. In older adults, a decrease in hippocampal volume has been observed, whereas healthy young women exhibit larger TIVs and larger volumes of total cortical grey and cerebral white matter [17].

The two forms of vitamin D are ergocalciferol (vitamin D_2_), found in plants, fungi, and yeast, and cholecalciferol (vitamin D_3_), derived from animal and vegetal sources. Vitamin D_3_ is mainly produced in the skin under the action of ultraviolet B (UVB) radiation [18]. Following the penetration of UVB rays (290–315 nm) into the epidermis, 7-dehydrocholesterol (7-DHC) is cleaved, leading to the formation of previtamin D_3_, which isomerises, forming vitamin D_3_, through a thermosensitive process [19]. Other sources of vitamin D are fortified foods and beverages [20].

Vitamin D is activated through a metabolic process that involves two hydroxylation reactions [21]. 25(OH)D, which serves as the main indicator of vitamin D status in the body, is synthesised in the liver, while 1,25-dihydroxyvitamin D (1,25(OH)_2_D), the active metabolite, is produced by hydroxylation of 25(OH)D in the kidneys [18]. 25(OH)D has a half-life of two to three weeks, whereas 1,25(OH)_2_D has a half-life of only four hours, being highly sensitive to fluctuations in parathyroid hormone (PTH) levels, which are triggered by subtle variations in calcium homeostasis [22]. The activity of vitamin D is closely related to its metabolism, as illustrated in Figure 1.

The mechanism by which vitamin D exerts its biological effects involves its binding to specific vitamin D receptors (VDRs). As a result of this interaction, the transcription of 3–5% of human genes is regulated [18]. Regardless of whether we refer to its direct action at the level of specific receptors or its indirect effects, vitamin D exerts its biological activity as a hormone by regulating more than 200 genes [23].

Taking into account that inflammation is one of the most frequently involved mechanisms in the pathophysiology of depression, and scientific evidence suggests a neuroimmunomodulatory effect of vitamin D, its use in relation to depressive disorder indicates a possible but yet-unconfirmed role in prevention and treatment strategies. Therefore, in this study, our objective was to review the relationship between serum 25(OH)D levels and depression, and to evaluate the effect of vitamin D supplementation on depressive symptoms.

## 2. Methods

The systematic review was carried out following the Preferred Reporting Items for Systematic Reviews and Meta-Analysis (PRISMA) guidelines.

### 2.1. Study Eligibility, Inclusion Criteria, and Exclusion Criteria

Relevant articles, based on inclusion and exclusion criteria, were selected for this systematic review, both in descriptive form within the corresponding chapters and in a tabular format.

The inclusion criteria were formulated as follows:Studies conducted on adults aged 18 years or older;Observational studies (such as cross-sectional, cohort, and case–control studies) and interventional studies (RCTs);Studies that assessed the relationship between vitamin D and depression;Studies that evaluated the effect of vitamin D supplementation in relation to depressive symptomatology;Studies in which vitamin D levels were determined by measuring serum 25(OH)D concentrations;Studies employing validated instruments to quantify depressive symptomatology;Articles published in English.

The exclusion criteria were formulated as follows:Studies that included participants under the age of 18;Studies conducted on animal models;In vitro studies;Articles published in a language other than English;Studies for which the full text was not available.

### 2.2. Search Strategy and Study Selection

To achieve the proposed objective, we conducted a report including a selection of scientific evidence necessary to review the relationship between vitamin D and depression. In this regard, using the keywords ‘depression’, ‘major depressive disorder’, ‘vitamin D’, ‘25-hydroxyvitamin D’, and their combinations, a search was conducted on the following platforms: PubMed, MDPI, ResearchGate, Springer Link, SpringerOpen, and ScienceDirect. Constraints were applied regarding the article type, language, and publication date. Research articles, published in English between 2008 and 2024, were selected. After excluding articles conducted on animal models and in vitro studies, a total of 13,976 records were obtained. Duplicate articles, amounting to 5774 across all databases, were excluded from the data set. Following the assessment of the title and abstract, a total of 8086 articles did not meet the criteria set for this study, and were excluded. After excluding studies without open access, a remaining 93 articles were thoroughly examined. Based on the inclusion and exclusion criteria, 70 articles were selected and included in this systematic review. The study selection process was carried out according to the PRISMA 2020 flow diagram shown in Figure 2.

The search strategy and study selection processes were performed independently by a single researcher.

### 2.3. Data Extraction

All the necessary information was available in the content of the articles included in the systematic review. The extracted information was related to the following: the study design (cross-sectional, cohort, observational studies, randomised clinical trials), the characteristics of the study population (age, gender, number of participants, nationality/country, diagnosis), the monitored parameters (measurement of serum vitamin D concentration, vitamin D supplementation, scales applied for mental health assessment, follow-up duration, type of intervention), and the results obtained.

The data extraction process was performed independently by a single researcher.

## 3. Results and Discussion

This section is structured into three main segments. The first part addresses the implication of vitamin D in the pathophysiological mechanism of MDD. The second part focuses on the relationship between serum 25(OH)D levels and depression. This correlation is analysed through cross-sectional and cohort studies. The third and final part includes studies evaluating the efficacy of vitamin D supplementation in relation to MDD. All data within is presented both in descriptive analysis and tabular form. The last two segments include positive and also null findings.

### 3.1. The Implication of Vitamin D in the Pathophysiology of Depression

The enzymes 25-hydroxylase and 25D-1α-hydroxylase, responsible for metabolising vitamin D into its active metabolite, have been identified in the central nervous system (CNS), suggesting the brain’s capacity to locally activate vitamin D [16,24]. Furthermore, the metabolite 1,25(OH)_2_D is inactivated in the brain by the action of the cytochrome P450 (CYP450) enzyme, CYP24A1. The brain’s ability to synthesise and eliminate the active metabolite suggests the involvement of autocrine and paracrine pathways in vitamin D signalling [25]. VDRs, through which the active metabolite exerts its action, are located in various regions of the brain, including the hippocampus. Their location, as well as the fact that the hippocampus presents modifications in patients with chronic depression, suggest the possible implications of vitamin D in the aetiology of MDD [26,27,28].

Although the effects of vitamin D on the brain are not yet fully understood, its action is mediated through various mechanisms that could support the theoretical relationship between vitamin D and the pathophysiology of depression [29].

Numerous studies have highlighted a connection between inflammation and depression, suggesting the involvement of additional mechanisms in the pathogenesis of MDD. Approximately 30% of patients with depression have been observed to exhibit a disruption of the inflammatory process, characterised by an increase in the concentration of inflammatory markers [9]. Persistently elevated levels of inflammatory cytokines disrupt the metabolism of serotonin, norepinephrine, and dopamine, leading to the stimulation of the hypothalamus–pituitary–adrenal (HPA) axis [30]. Abnormalities within the axis, such as excessive glucocorticoid secretion, resistance to the glucocorticoid receptors (GRs), and impaired negative feedback, may be present in MDD patients [31].

Vitamin D has been shown to play an essential role in the chronic inflammatory process, influencing the function of T cells and various cytokines [32]. Its anti-inflammatory effect is manifested through an increase in anti-inflammatory cytokine levels, such as interleukin-10 (IL-10), interleukin-4 (IL-4), and interleukin-5 (IL-5); a reduction in the concentration of pro-inflammatory cytokines, including interleukin-1 beta (IL-1β), interleukin-2 (IL-2), interleukin-6 (IL-6), interleukin-12 (IL-12), interferon-gamma (INF-γ), and tumour necrosis factor-alpha (TNF-α) [33]; and inhibition of mitogen-activated protein kinase (MAPK) and nuclear factor kappa-light-chain-enhancer of activated B cells (NF-kB) [34]. Considering the relationship between VDRs and GRs in the hippocampus, vitamin D can antagonise various effects mediated by glucocorticoids [29].

As previously mentioned, the monoaminergic theory proposes that deficiency in monoaminergic neurotransmissions (serotoninergic, noradrenergic, and dopaminergic) is the primary mechanism involved in the onset of depression [8]. Vitamin D induces the activity of the enzyme tryptophan hydrolysis 2 (TPH2), stimulating tryptophan hydroxylation and increasing serotonin synthesis [35], and regulates the expression of the serotonin reuptake transporter (SERT) and the level of monoamine oxidase A (MAO-A), thus influencing the serotonin concentration in the brain [19]. Through the activation of gene expression of the enzyme tyrosine hydroxylase, vitamin D influences the levels of dopamine, epinephrine, and norepinephrine. Through induction of choline acetyltransferase activity and inhibition of acetylcholine esterase activity, vitamin D may modulate cholinergic function [36]. Additionally, it can regulate the synthesis of monoaminergic neurotransmitters in the adrenal cortex as a result of the modulation of the HPA axis [37].

The neuroprotective effect of vitamin D is exerted through various mechanisms, such as the enhancement of the antioxidant defence system [18], the modulation of neurotrophic factors [36], and the regulation of intracellular calcium metabolism [38]. Firstly, vitamin D influences the synthesis of glutathione, one of the most important antioxidant agents in the brain, by modulating the enzyme gamma-glutamyl transpeptidase [39]. Secondly, it stimulates the activity of nerve growth factor (NGF), glial cell line-derived neurotrophic factor (GDNF), and neurotrophin 3 (NT-3), and decreases the activity of neurotrophin 4 (NT-4) [27]. Last but not least, vitamin D has been observed to regulate, through various mechanisms, the increased Ca^+2^ concentration at the intraneuronal level, which has been proposed as a potential cellular cause of depression [12]. For example, vitamin D is believed to increase the expression of the calbindin and parvalbumin genes, thereby promoting the conversion of Ca^+2^ to buffered forms. It has also been observed to induce the expression of plasma membrane Ca^+2^-ATPase (PMCA) and Na^+^/Ca^+2^ exchanger 1 (NCX1), thus facilitating the elimination of excess Ca^+2^. Another hypothesis has suggested its capacity to decrease the expression of the L-type CaV1.3 and CaV1.2 channels in the hippocampal and cortical neurones [40].

Regulation of PTH homeostasis is another possible mechanism through which vitamin D may influence the aetiology of depression. Numerous studies have suggested that a decrease in serum 25(OH)D concentration is correlated with an increase in PTH levels, and these elevated values are associated with depressive disorders [41,42].

The antidepressant-like properties of vitamin D, particularly those targeting the inflammatory mechanism, have also been examined in preclinical animal studies [31]. For example, a study using ovariectomised female Sprague Dawley rats demonstrated that administration of calcitriol led to the regulation of the AMP-activated protein kinase (AMPK)/NF-kB signalling pathway and a reduction in the expression of inducible nitric oxide synthase (iNOS), cyclooxygenase-2 (COX-2), and pro-inflammatory cytokines (IL-1β, IL-6, TNF-α) [43].

Figure 3 illustrates the relationship between the mechanisms of action of vitamin D and the pathogenesis of depressive disorder.

The anti-inflammatory effect, induction of tryptophan hydroxylase 2 activity, enhancement of the antioxidant defence system, influence on neurotrophic factors, regulation of intracellular calcium metabolism, and control of parathyroid hormone homeostasis are some of the mechanisms through which vitamin D may influence the pathophysiology of depression. The involvement of vitamin D in the processes that occur within the CNS, as well as the presence of the enzymes and VDRs, suggest a potential correlation between vitamin D and MDD.

### 3.2. The Relationship Between Serum Vitamin D Levels and Depression

The concentration of the vitamin D metabolite is considered optimal within the range of 30 to 60 ng/mL, while values below 20 ng/mL are classified as deficient, and those between 21 and 29 ng/mL are categorised as insufficient [44]. Considering that vitamin D levels are maintained both through endogenous production after exposure to sunlight and through dietary sources, determining the daily requirement can be a more complex process [45]. Depending on age, the recommended daily intake according to guidelines is at least 400 IU for infants (0–12 months); 600 IU for children (1–13 years), adolescents (14–18 years), and adults (19–70 years); and 800 IU for individuals over 70 years of age [46]. From the diet, the average intake is approximately 347.05 ± 307.8 IU, which is significantly below the recommended daily requirement, while cutaneous production depends on the degree of exposure to the sun and the factors that influence this [47].

Hypovitaminosis D is a public health concern, with estimates indicating that worldwide, approximately 30% of children and 60% of adults have insufficient or deficient levels of vitamin D [48]. Low levels of 25(OH)D have been correlated with various metabolic disorders, such as obesity, diabetes mellitus, insulin resistance, and hypertension [49], as well as with neuropsychiatric disorders and neurodegenerative conditions, including schizophrenia, Parkinson’s disease, Alzheimer’s disease, cognitive impairment, and depression [34].

There are many factors that can negatively influence the serum concentration of the vitamin D metabolite. Clothing, the use of sunscreens, geographic location, and the season are just a few of the physical factors that ultimately influence vitamin D synthesis in the skin. Among biological factors are the degree of cutaneous pigmentation, body mass index (BMI), age, and the genetic polymorphism of receptors and enzymes involved in metabolisation [18,41]. Genes that can influence serum vitamin D concentration include those specific to the CYP2R1 enzyme, which is responsible for the conversion of vitamin D to 25(OH)D in the liver, and the CYP24A1 enzyme, involved in the transformation of 25(OH)D into 24,24-dihydroxyvitamin D. Additionally, variations in the group-specific component (GC) gene, which encodes vitamin D-binding protein (VDBP), as well as the gene involved in 7-DHC synthesis in the skin, may contribute to differences in vitamin D levels within the body [50]. Gastrointestinal disorders, such as Crohn’s disease (CD), celiac disease, malabsorption syndrome, lactose intolerance, and short bowel syndrome, can condition the degree of absorption from dietary sources and supplements. Among the category of drugs that can regulate vitamin D metabolism and action, we mention glucocorticoids, bisphosphonates, antioestrogens, antituberculosis drugs, HMG-CoA-reductase inhibitor drugs, and cytostatic agents. The state of liver and kidney function, as well as certain conditions, such as hyperparathyroidism, sarcoidosis, tuberculosis, histoplasmosis, and lymphatic cancer, can influence serum 25(OH)D concentration by reducing its absorption [7]. MDD may contribute to the onset of vitamin D deficiency due to its impact on lifestyle, physical activity, diet, and body weight [51].

Numerous studies have suggested a correlation between serum vitamin D levels and the presence of depressive symptoms [42,52], with its deficiency associated with an 8–14% increase in the risk of depression [53]. The relationship between serum 25(OH)D concentration and the risk of depression has been investigated in various studies, with the results indicating that the level of the metabolite is inversely correlated with the risk of depression [26,27,37,54]. The incidence of depression was found to be approximately three times higher in individuals with very low 25(OH)D levels (≤15 ng/mL) and twice as high in those with low or normal levels [41].

Scientific evidence suggests that depressive subjects have lower vitamin D levels when compared to a control group, with those exhibiting the lowest levels being at a higher risk of depression [37]. This hypothesis was highlighted by Okasha et al. in a sample consisting of 20 patients with MDD, 20 patients with schizophrenia, and 20 healthy participants. Compared to the control group, patients diagnosed with depression had a significantly lower level of vitamin D [55].

From a population perspective, in the Nepalese population [56], Finnish men [57], Japanese workers [58], Jordanian adults [59], and European men [60], an inverse relationship between serum vitamin D levels and the odds [56,60], risk [57,59], and prevalence [58] of depression has been observed.

Fifty patients diagnosed with MDD were included in a study aimed at measuring serum vitamin D concentrations and calculating the TIV using high-resolution structural magnetic resonance imaging (HRMRI). The findings indicated that depressive symptoms were more severe in patients with lower serum levels of 25(OH)D and a reduced TIV [17]. The same researchers conducted a study in which they investigated the relationship between gender, vitamin D, clinical manifestations, and functional connectivity of brain networks in a sample comprising MDD patients and healthy subjects. The results obtained highlighted that hypovitaminosis D was more prevalent among women than men, and that female patients with depression exhibited significantly lower levels of vitamin D compared to the control group. A possible association between low vitamin D levels, functional brain network dysfunction, and clinical manifestations was observed in female subjects diagnosed with MDD [61].

In a cross-sectional study involving 3926 adults from the general population, an inverse relationship was identified between vitamin D and depression, as well as between vitamin D and obesity [62]. A comparative observational study, conducted on obese subjects between 18 and 60 years of age, has highlighted a correlation between low serum 25(OH)D levels and the incidence of depression. Hypovitaminosis D was significantly more prevalent in depressive patients (78%) compared to individuals without depression (67%). Insufficient dietary intake, reduced intestinal absorption, increased adipose tissue and muscle mass, decreased cutaneous synthesis, altered metabolism, and limited sun exposure due to low physical activity are some of the factors contributing to the significantly higher prevalence—by 35%—of hypovitaminosis among overweight individuals compared to those with normal weight. This negative correlation between obesity and vitamin D, as well as the fact that both contribute to the onset of a chronic inflammatory process, may play a role in the development of depression [42].

Insufficient dietary intake, limited sun exposure, reduced renal capacity to produce calcitriol, or malabsorption caused by gastrointestinal disorders or the use of certain medications are among the key factors underlying the high prevalence of vitamin D deficiency in the elderly population [63]. Studies conducted on these individuals have suggested an inverse association between vitamin D levels and both the prevalence of depression [29,64] and the severity of depressive symptoms [65,66]. Even after controlling for confounding factors, low serum levels of 25(OH)D were associated with higher scores on depression scales, indicating a worsening of symptomatology [65].

In two studies, one conducted on patients diagnosed with rheumatoid arthritis (RA) [67] and the second on patients with chronic liver disease (CLD) [68], the researchers confirmed a significant inverse correlation between serum vitamin D levels and the severity of depressive symptoms [67,68].

Postpartum depression (PPD) is a common affective disorder that affects approximately 20–40% of women worldwide after childbirth, with detrimental consequences for the mother–child relationship during the first 12 months after delivery, as well as for the child’s emotional and cognitive development. The correlation between hypovitaminosis D and PPD is not yet fully understood, as existing evidence remains contradictory. However, cohort studies indicate an association between vitamin D deficiency and the incidence of depression, suggesting a potential role of vitamin D in the recovery of women affected by PPD [69]. The results of a systematic review and meta-analysis, which included 15 cohort studies, 9 cross-sectional studies, and 1 case–control study, highlighted an inverse relationship between serum 25(OH)D concentration and the presence of depressive symptoms during the antenatal and postnatal periods [70].

A cross-sectional study conducted on pregnant women, postpartum women, non-pregnant/postpartum women, and men indicated that lower vitamin D concentrations may be associated with a higher likelihood of depression [71].

Table 1 presents cross-sectional studies that have highlighted the relationship between serum vitamin D levels and depression.

However, there are also studies in which an association between serum levels of the vitamin D metabolite and depression could not be demonstrated. The majority of studies analysing this relationship are observational, while the number of RCTs, which are considered superior for establishing causality, is limited [37]. Heterogeneity among studies may result from their quality (such as a small sample size) or from a lack of control over certain variables, such as gender, BMI, genetic characteristics, socioeconomic status, diet, alcohol consumption, or smoking, which may lead to discrepancies in the results obtained [27].

In a study conducted on 5006 older adults (66–96 years) living in communities in northern regions, a modest association was observed between serum metabolite concentration and depression [73]. Zhao et al. reported no association after adjusting for confounding factors (demographic variables, lifestyle-related factors, and the presence of chronic conditions) in a sample that included 3916 participants aged 20 and over [75]. A study in China involving 3262 subjects aged 50 to 70 years revealed that the association between 25(OH)D levels and the prevalence of depressive symptoms was attenuated after adjustment for various confounding factors, and disappeared entirely when geographic location was included in the analysis [76]. Another study involving 63 overweight or obese adults (39 men and 24 women) with hypovitaminosis D and no clinical depression found no association between serum 25(OH)D levels and depressive symptoms [77]. In a sample consisting of 489 menopausal women, no association was observed between depression and vitamin D levels [78]. An analysis of data from three studies involving 5308 Danish adults found no relationship between serum 25(OH)D concentrations and depressive symptoms, even after adjusting for confounding factors [79].

Table 2 presents cross-sectional studies that have not highlighted the relationship between serum vitamin D levels and depression.

From the perspective of longitudinal studies that examine the long-term correlation between vitamin D and depression, the results are inconsistent. In a cohort study, this relationship was investigated in a sample of 1102 patients with current depression, 790 patients with remitted depression, and 494 healthy subjects. After controlling for covariates, low levels of the 25(OH)D metabolite were found to be associated with the presence and severity of depressive symptoms, and vitamin D deficiency and insufficiency were found to be more frequent among subjects with current or remitted depression. Furthermore, the progression of patients with current depression and low levels of 25(OH)D was less favourable over a two-year period. The authors suggested that hypovitaminosis D may represent an underlying biological vulnerability in the pathogenesis of depression [51]. After a period of monitoring the relationship between vitamin D and the onset of depression, the results of a study revealed that participants with low serum levels of 25(OH)D had a higher risk of developing depressive symptoms over a period of six years [80]. Briggs et al. demonstrated that hypovitaminosis D was associated with a 75% increased risk of developing depression over a four-year period in a sample of elderly individuals without pre-existing depression, even after controlling for variables (such as physical activity, chronic illnesses, and CV conditions) and excluding subjects who were using antidepressant medications or vitamin D [81].

Table 3 presents cohort studies that have highlighted the relationship between serum vitamin D levels and depression.

Although a cross-sectional association has been observed between vitamin D metabolite concentration and depression, low serum levels of 25(OH)D were not prospectively correlated with changes in depressive symptoms or with the incidence of affective disorders after follow-up periods of four years [72] and ten years [74], respectively. In a sample comprising 232 elderly patients with depression, vitamin D levels decreased over a two-year period compared to the initial assessment, independently of the course of depression [82].

Table 4 presents cohort studies that have not highlighted the relationship between serum vitamin D levels and depression.

The duration of monitoring, the method used to measure the serum vitamin D concentration, confounding factors, and the scales employed to assess psychological function are some of the methodological differences found in longitudinal studies that influence the results and the conclusions drawn [72,81]. For these reasons, well-designed, large-scale prospective cohort studies are necessary to provide a definitive answer regarding the relationship between vitamin D and depression [74].

In summary, this subsection includes both cross-sectional and cohort studies that evaluated the relationship between serum 25(OH)D levels and depression in heterogeneous samples. While some studies observed an inverse correlation between these two parameters, others showed no significant association.

### 3.3. The Role of Vitamin D Supplementation in Depressive Symptoms

The relationship between vitamin D supplementation and serum 25(OH)D concentration may be influenced by numerous environmental and demographic factors. For example, variations in metabolite concentration following supplementation may be determined by body weight (34.5%), the type of supplement (9.8%), age (3.7%), calcium intake (2.4%), and baseline 25(OH)D levels (1.9%). A high BMI, particularly above 30 kg/m^2^, has been associated with smaller increases in serum 25(OH)D concentration after supplementation, possibly due to the fat-soluble nature of vitamin D, which promotes its storage in adipose tissue. Another relevant factor is age, as advancing age is frequently linked to hypovitaminosis D, primarily due to a decline in the levels of 7-DHC. Supplementation is more beneficial in individuals with a deficiency, likely due to the fact that hepatic hydroxylation is a saturable process [18].

Information obtained from the specialised literature indicates an improvement in depressive symptomatology, as evidenced by enhanced scores on depression assessments, following an increase in serum 25(OH)D concentration [2,83,84]. Scientific evidence indicates that each 10 ng/mL increase in metabolite levels is associated with a 12% reduction in the risk of depression [26].

Several studies have focused on the efficacy of vitamin D administered as monotherapy in the management of depressive symptoms. For example, two double-blind, randomised clinical trials converged toward similar conclusions. Administration of vitamin D_3_ (50,000 IU every two weeks) over a period of eight weeks led to a significant increase in serum 25(OH)D levels in subjects with mild-to-moderate depression in the experimental group and significantly improved the severity of depressive symptoms [12,85]. In a randomised clinical trial involving 120 patients with depression and hypovitaminosis D, a single intramuscular dose of 300,000 IU or 150,000 IU of vitamin D was administered. Vitamin D deficiency is highly prevalent among patients with depression, and restoring its levels leads to improvement in depressive symptoms. Higher doses, such as 300,000 IU compared to 150,000 IU, are more efficient and also safe [86].

Although most researchers have evaluated the isolated effect of vitamin D supplementation compared to a placebo, in recent years, the focus has also shifted toward its role as an adjunct to antidepressant treatment [44]. In this regard, Vellekkatt et al. conducted a randomised, double-blind, placebo-controlled study aimed at evaluating the efficacy of a single dose of vitamin D_3_, administered at baseline, as an adjunct therapy to standard treatment. The results indicated that the administration of a single dose of vitamin D_3_ had a positive impact on both depressive symptoms and quality of life, thus demonstrating the efficacy of the intervention in subjects with MDD and hypovitaminosis D [87]. An 8-week study carried out on 78 older adults with moderate-to-severe depression found that vitamin D_3_ supplementation (50,000 IU/week) may serve as an effective adjunctive therapeutic strategy. Compared to the control group, which received a placebo in conjunction with standard antidepressant treatment, the experimental group exhibited a significant improvement in depression scores [88]. Khoraminya et al. conducted a randomised, placebo-controlled clinical trial to evaluate the efficacy of adding vitamin D to standard fluoxetine treatment in depression. The study results indicated a synergistic relationship between vitamin D and fluoxetine, leading to a significantly enhanced treatment response compared to fluoxetine monotherapy [89]. A study by Alghamdi et al. investigated the impact of vitamin D_3_ supplementation over a three-month period in 62 patients diagnosed with MDD, who were divided into two groups. The first group received vitamin D_3_ (50,000 IU/week) in addition to standard treatment, while the second group received standard treatment only. When analysing gender differences, it was found that women diagnosed with moderate, severe, or extreme depression showed lower depression scale scores after vitamin D use, while among men, only those with severe depression exhibited an improvement [49].

MDD does not always occur in isolation, but is often a component of complex syndromes in which it coexists with other chronic conditions. For this reason, researchers have conducted studies investigating the effect of vitamin D on depressive symptoms in patients with certain associated comorbidities, such as type 2 diabetes mellitus (T2DM) [90,91], diabetic peripheral neuropathy (DPN) [92], CLD [68], some inflammatory gastrointestinal disorders [35,93,94], and inflammatory joint diseases [95]. The results highlight the superiority of vitamin D administration in reducing depressive symptoms, even in the context of an affective disorder overlapping with other comorbidities [35,68,90,91,92,93,94,95]. Zaromytidou et al. investigated the effect of vitamin D supplementation on anxiety and depression in elderly individuals with prediabetes. Participants with vitamin D deficiency and insufficiency at baseline experienced similar benefits in terms of anxiety and depression scores following supplementation [96].

The specialised literature highlights the role of methadone maintenance treatment in the management of opioid dependence. Given the high prevalence of this issue in Iran, where approximately 1.2 million people are affected, and the association between methadone and mental health parameters, such as sleep, anxiety, and affective disorders, it has become necessary to identify new strategies to modulate the effectiveness of methadone treatment. Thus, Ghaderi et al. explored the effect of vitamin D supplementation through two randomised, double-blind, placebo-controlled clinical trials, and showed the benefits of its administration on cognitive functions and certain mental health parameters [97,98].

Vaziri et al. conducted a randomised clinical trial to evaluate the impact of vitamin D supplementation on reducing perinatal depressive symptoms. The study results indicated that the administration of 2000 IU of vitamin D_3_ daily during the third trimester of pregnancy was associated with a significant improvement in depression scores during the perinatal period. This finding supports the hypothesis that vitamin D may contribute to the pathogenesis of perinatal depression [16]. Lv et al. carried out a study in which 800 IU of vitamin D per day was administered to a sample of 1365 pregnant women with vitamin D deficiency between the 12th and 14th weeks of gestation. The results showed a reduction in depressive symptoms among individuals with vitamin D deficiency at the beginning of pregnancy [99]. Another study highlighted the positive effect of vitamin D use from weeks 26 to 28 of gestation until birth on depression scores in the fourth postpartum week [100].

Several studies have investigated the synergy between vitamin D and other nutrients in the context of mental health. The effect of co-supplementation with vitamin D and omega-3 fatty acids on mental health parameters was examined in a sample of 60 women diagnosed with polycystic ovary syndrome [101]. Another study explored the impact of zinc, vitamin D, and their combined supplementation, compared to a placebo, on depressive symptoms in 140 obese or overweight patients diagnosed with MDD [102]. The effect of co-supplementation with probiotics and vitamin D_3_ was assessed in a study conducted on diabetic patients with coronary heart disease [103]. Amini et al. designed a scientific study through which they investigated the impact of combining vitamin D with calcium carbonate on the severity of symptoms in 27 women with PPD [104]. Abiri et al. explored the efficacy of combined administration of vitamin D with magnesium in a sample of 108 obese women with mild-to-moderate depressive disorder [5]. The results highlighted the beneficial effect of vitamin D and its combination with omega-3 fatty acids [101], zinc [102], probiotics [103], calcium carbonate [104], and magnesium [5] on depression scores, compared to groups receiving a placebo.

Table 5 presents studies that have demonstrated the effectiveness of vitamin D supplementation in relation to MDD.

Although RCTs are the most important for determining the efficacy of vitamin D supplementation in depression, some of them have produced inconsistent results and failed to demonstrate a significant improvement in depressive symptoms. The sample size, age range of the participants, vitamin D dosage, method of administration, scales used to measure the severity of depression, and outcome assessment are some examples of variables that may influence the results of these studies, making it difficult to generalise the findings [44]. Systematic review studies and meta-analyses including such articles have failed to confirm a positive effect on mental health parameters following vitamin D supplementation [48,105,106].

A randomised, placebo-controlled clinical trial conducted on a cohort of 21,315 Australian adults did not demonstrate an overall benefit of monthly supplementation with 60,000 IU of vitamin D_3_ over a period of five years. However, preliminary subgroup analyses suggested a potential effect among individuals who used antidepressant medication at baseline or who had lower baseline serum concentrations of 25(OH)D [107].

A randomised, double-blind, placebo-controlled clinical trial conducted over the course of one year aimed to investigate the impact of vitamin D supplementation, administered in varying doses, in elderly women from the general population [108]. This study included two works by Gallagher et al., in which 163 Caucasian women [45] and 110 African American women [50], aged between 57 and 90 years and with serum 25(OH)D levels ≤ 20 ng/mL, were recruited. The study involved the random allocation of subjects to one of eight groups, as follows: 400, 800, 1600, 2400, 3200, 4000, and 4800 IU of vitamin D_3_ per day, or a matching placebo. The results did not reveal a significant beneficial effect, possibly due to the relatively small number of depressed women within the groups [108]. The study by Choukri et al., involving 152 healthy premenopausal women, did not identify an impact of monthly administration of vitamin D [109].

Two studies conducted on overweight individuals, one involving postmenopausal women [110] and the other focusing on adults from the general population [77], showed no effect on depression scores after vitamin D supplementation [77,110].

Jorde et al. conducted a randomised study involving 408 participants, comparing the impact of vitamin D administration (a 100,000 IU bolus followed by 20,000 IU per week) with a placebo for a period of four months [111]. Hansen et al. evaluated the efficacy of vitamin D supplementation, compared to a placebo, along with standard antidepressant treatment, in a sample of 62 patients with mild-to-severe depression [11]. Marsh et al. conducted a study on patients diagnosed with bipolar depression and insufficient vitamin D levels [22]. Koning et al. carried out a study on 155 elderly individuals with hypovitaminosis D and clinically relevant depressive symptoms [112]. All four studies led to the same result, emphasising that vitamin D supplementation had no significant effect on depressive symptoms [11,22,111,112].

The effect of vitamin D supplementation has been evaluated in patients diagnosed with end-stage chronic kidney disease (CKD) undergoing dialysis [113], patients with multiple sclerosis (MS) [114], patients with pulmonary tuberculosis (PTB) [115], and patients with T2DM [116]. In all instances, the results did not highlight a significant beneficial effect of vitamin D on depressive symptoms [113,114,115,116].

The effect of vitamin D supplementation on the risk of depression and the long-term course of mood was examined in a sample of 18,353 middle-aged and older adults without depression at baseline. Compared to the placebo group, daily administration of 2000 IU of cholecalciferol combined with fish oil (1 g capsule per day containing 840 mg of omega-3 fatty acids, including 465 mg of eicosapentaenoic acid and 375 mg of docosahexaenoic acid) did not result in a statistically significant difference in the incidence and recurrence of depressive symptoms over a follow-up period of 5.3 years [117]. Neither the combination of hormone therapy (oestrogen for women with hysterectomy, or oestrogen plus medroxyprogesterone for those with an intact uterus) with calcitriol, nor their individual use, yielded significant improvements in depressive symptoms in a sample of 489 postmenopausal women [78].

Table 6 presents studies that have not demonstrated the effectiveness of vitamin D supplementation in relation to MDD.

These controversies may reflect certain limitations, such as those related to study design, which affect the ability to demonstrate a change in vitamin D status within the intervention group. The concept of ‘biological flaws’, which defines these limitations, refers to inadequate interventions (those that did not include vitamin D), interventions that produced the opposite of the intended effect (included vitamin D, but reduced metabolite levels in the intervention group), ineffective interventions that failed to improve vitamin D status (did not significantly alter metabolite levels), and instances where baseline 25(OH)D levels were not measured in most participants or were already sufficient at the start of the study. In the conclusion of the systematic review conducted by Spedding et al., which compared studies with and without ‘biological flaws’, the researchers emphasise that all studies without such flaws—and their corresponding meta-analysis—supported the efficacy of vitamin D supplementation in MDD [118].

The subgroup analysis conducted in the study by Shaffer et al. demonstrated that vitamin D supplementation had a moderate and statistically significant effect only in the sample that included patients with depressive symptoms. Among participants without depression, the results indicated a nonsignificant impact [119]. Higher doses (≥4000 UI) have a more favourable effect on depressive symptomatology [2], with the effect being more pronounced in individuals with low levels of 25(OH)D [48,120]. Furthermore, studies have suggested that the most effective supplementation regimen requires a duration of at least 8 weeks [121], and noticeable improvement in depressive symptoms is observed in follow-up periods shorter than 24 weeks [6]. The results of a meta-analysis demonstrated that vitamin D supplementation influenced both the incidence and the course of depression [120].

To summarise, this subsection included interventional studies that evaluated the impact of vitamin D supplementation in relation to depression, and were conducted on clinically diverse cohorts. The evidence is inconsistent, with positive findings observed in some studies, and no effect in others.

This systematic review is among the few existing works that provide a comprehensive and up-to-date summary of the scientific evidence highlighted by the specialised articles that have followed the implications of vitamin D in the management of MDD, both in terms of the link between the two parameters and in terms of the effect of vitamin D administration in relation to depression. In this regard, the article included cross-sectional studies examining the correlation between serum 25(OH)D concentration and depression at a given time, and cohort studies following the two parameters in the long term, in order to establish causality. To understand the impact of vitamin D supplementation on depressive symptoms, RCTs were incorporated, which are the most relevant in terms of determining effectiveness. In order to control as much as possible the concept of ‘biological flaws’, which defines the limitations related to the design of the studies, only articles in which the value of vitamin D was determined by dosing the serum concentration of 25(OH)D, and in which validated instruments were used to quantify depressive symptoms, were selected. Another strong point of the present paper is represented by the chapter in which some of the effects of vitamin D proposed to be involved in the pathophysiological mechanism of MDD are described. The systematic review encompasses a diverse spectrum of articles, including recently published articles, thus allowing previous findings to be updated.

This increased number of included studies can be seen as a strength, but also as a limitation, due to the wide variation in study methodologies, which could make it difficult to generalise the results obtained. Discrepancies related to the results obtained may be the consequence of the quality of the articles or the lack of control of some confounding factors, which differ from one study to another and can lead to inconsistent results. The heterogeneity of the articles excluded the possibility of conducting a meta-analysis, but this shortcoming is compensated for by the value of the information, which contributes significantly to completing and updating the existing knowledge in the field, providing a useful basis for future research.

## 4. Conclusions

Given the limitations of conventional therapies, including alternative and adjunctive treatments, the identification of additional therapeutic options could open up new perspectives in the prevention and management of MDD.

Evidence from the scientific literature suggests the beneficial potential of vitamin D supplementation both as a prophylactic strategy to reduce the risk of developing depression and as an adjunctive therapy for symptom control. This hypothesis is supported by scientific findings highlighting the involvement of vitamin D in neurobiological mechanisms associated with depression, as well as the negative correlation between serum 25(OH)D levels and depression prevalence.

However, there are cross-sectional and cohort studies that have failed to evidentiate an inverse relationship between serum 25(OH)D concentration and depression, and RCTs that have not highlighted a significant beneficial effect of vitamin D supplementation on depressive symptoms. The current level of evidence from the scientific literature suggests that vitamin D administration could be effective in some limited conditions. Its utilisation in patients with depressive symptoms and low levels of 25(OH)D, in higher doses and over a period of at least eight weeks, indicates an improvement in mental health status.

This systematic review also aims to raise awareness about the impact of vitamin D on mental health, and to highlight the importance of the topic for future research. In conclusion, high-quality studies, which are free from methodological errors, are needed to further investigate the relationship between serum 25(OH)D levels and the risk of depression, and to accurately assess the effectiveness of vitamin D supplementation on depressive symptoms.

## Figures and Tables

**Figure 1 pharmaceuticals-18-00792-f001:**
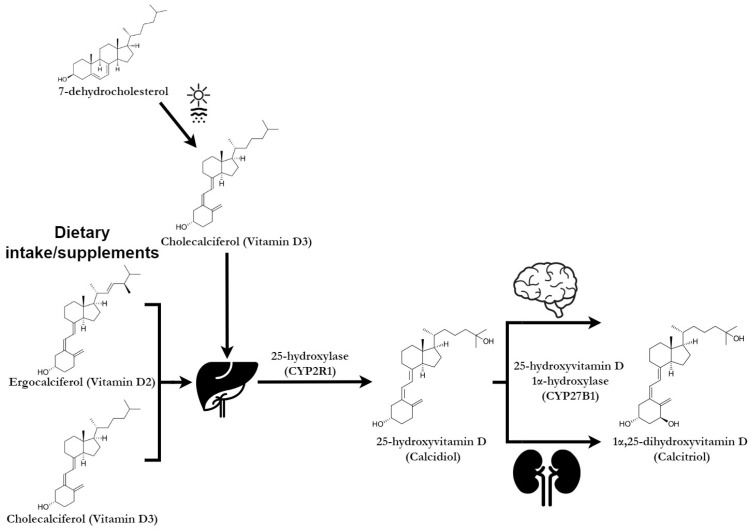
Vitamin D metabolism.

**Figure 2 pharmaceuticals-18-00792-f002:**
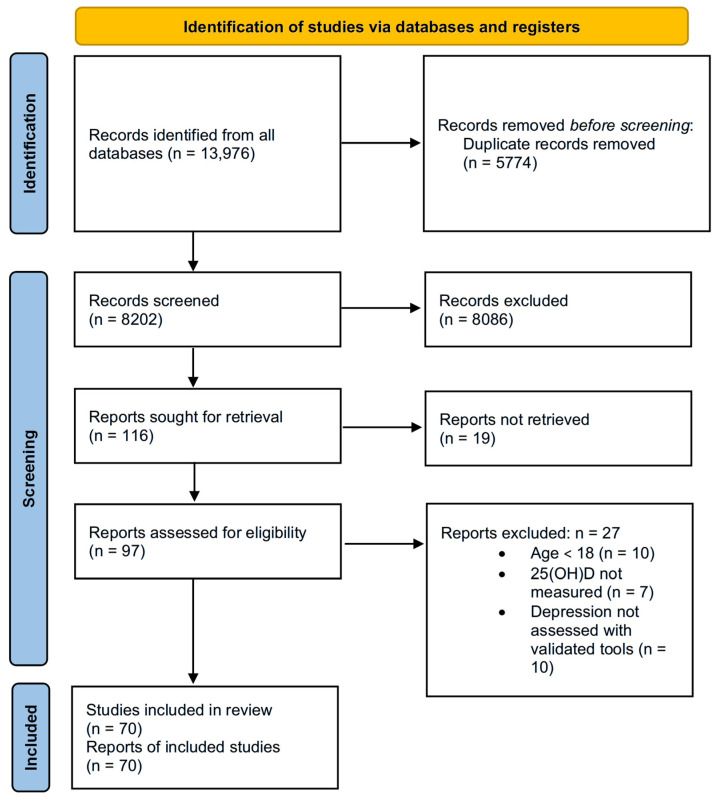
PRISMA 2020 flow diagram for searches of databases and registers. 25(OH)D = 25-Hydroxyvitamin D.

**Figure 3 pharmaceuticals-18-00792-f003:**
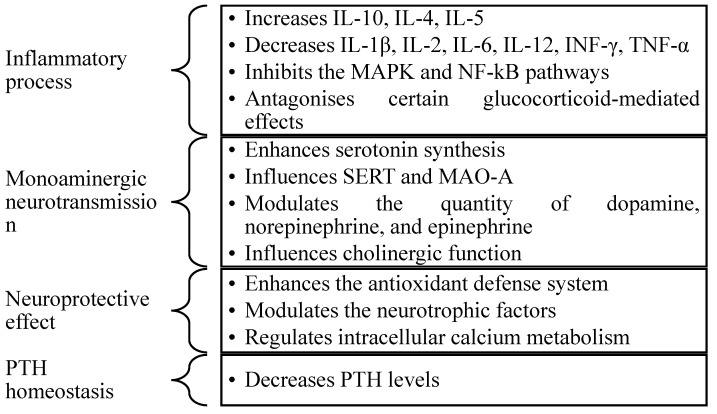
The relationship between the mechanisms of action of vitamin D and the pathogenesis of depressive disorder. INF-γ = interferon-gamma; IL-10 = interleukin-10; IL-4 = interleukin-4; IL-5 = interleukin-5; IL-1β = interleukin-1 beta; IL-2 = interleukin-2; IL-6 = interleukin-6; IL-12 = interleukin-12; MAO-A = monoamine oxidase A; MAPK = mitogen-activated protein kinase; NF-kB = nuclear factor kappa-light-chain-enhancer of activated B cells; SERT = serotonin reuptake transporter; PTH = parathyroid hormone; TNF-α = tumour necrosis factor-alpha.

**Table 1 pharmaceuticals-18-00792-t001:** Cross-sectional studies that have highlighted the relationship between serum vitamin D levels and depression.

First Author, Year	Design of Study	Study Population (N)	Age (Years)	Depression Assessment Scale	Results	Ref.
Brouwer-Brolsma et al., 2016	Cross-sectional	Dutch older adults (2839)	≥65	GDS	The study confirmed the presence of an inverse correlation between serum vitamin D levels and the severity of depressive symptoms.	[66]
Chan et al., 2011	Cross-sectional	Community-dwelling Chinese men (939)	>65	GDS	At the beginning of the study, a negative correlation was identified between the level of the vitamin D metabolite and depression.	[72]
Goltz et al., 2017	Cross-sectional	Adults from general population (3926)	20–79	PHQ-9	Serum 25(OH)D concentration was negatively associated with depression and obesity.	[62]
Hollinshead et al., 2024	Cross-sectional	Pregnant women, postpartum women, non-pregnant/postpartum women, and men (11,337)	20–44	PHQ-9	The study highlighted the presence of an inverse relationship between depressive symptomatology and vitamin D levels.	[71]
Hoogendijk et al., 2008	Large population-based study	Community-dwelling older people (1282)	65–95	CES-D	The study revealed a strong association of lower serum 25(OH)D levels and higher PTH levels with both the presence and severity of depression.	[65]
Imai et al., 2015	Cross-sectional	Community-dwelling older people living in Iceland (5006)	66–96	GDS-15; DSM-IV	Low serum 25(OH)D levels were modestly associated with higher depressive symptom scores.	[73]
Jaddou et al., 2012	Cross-sectional	Jordanian adults (4002)	≥25	DASS21	The study highlighted an inverse association between serum 25(OH)D concentrations and the prevalence of depression.	[59]
Jovanova et. al., 2017	Cross-sectional	Older adults (3251)	≥55	CES-D	The study highlighted the presence of an inverse relationship between the level of the vitamin D metabolite and depressive symptoms, from a cross-sectional perspective.	[74]
Jääskeläinen et al., 2015	Cross-sectional	Finnish men and women from the Health 2000 Survey (5371)	30–79	BDI	The risk of depression was negatively associated with vitamin D concentration.	[57]
Kamalzadeh et al., 2021	Comparative observational study	Obese individuals with depression (174) and without depression (173)	18–60	DSM-V	The results suggested the presence of a negative relationship between vitamin D levels and the incidence of depression in obese individuals.	[42]
Lee et al., 2011	Cross-sectional	Middle-aged and older men (3369)	40–79	BDI-II	The study highlighted an association between low serum 25(OH)D levels and depression, a relationship that persisted even after controlling for certain covariates.	[60]
Mizoue et al., 2015	Cross-sectional	Healthy Japanese workers (1786)	19–69	CES-D	The results showed that subjects with low vitamin D levels exhibited an increased likelihood of experiencing depressive symptoms.	[58]
Okasha et al., 2020	Cross-sectional	Patients with MDD (20), patients with schizophrenia (20), and healthy control subjects (20)	20–50	SCID-I	Patients diagnosed with MDD and schizophrenia exhibited low concentrations of 25(OH)D.	[55]
Pu et al., 2018	Cross-sectional	Rheumatoid arthritis patients (161)	25–75	HAMD; HAMA	Lower serum concentrations of the vitamin D metabolite were identified in subjects with depression or anxiety.	[67]
Sherchand et al., 2018	Cross-sectional	Adults residing in eastern Nepal (300)	≥18	BDI-Ia (validated Nepali version)	The results identified the presence of an inverse relationship between vitamin D deficiency and the likelihood of clinically significant depression.	[56]
Stewart et al., 2010	Cross-sectional	Older people who had participated in the 2005 Health Survey for England (2070)	≥65	GDS	The study highlighted a negative relationship between serum 25(OH)D levels and late-life depression.	[64]
Stokes et al., 2016	Cross-sectional	CLD patients (111)	20–81	BDI-II	A negative relationship was observed between the severity of depressive symptoms and serum vitamin D levels.	[68]
Vidgren et al., 2018	Cross-sectional	General middle-aged or older population (1602)	53–73	DSM-III	The study highlighted the presence of an inverse relationship between serum vitamin D levels and the prevalence of depression in older adults.	[29]
Zhu et al., 2019	Cross-sectional	Patients with MDD (50)	18–60	HAMD; HAMA	The study suggested that low serum concentration of 25(OH)D is associated with increased severity of depressive symptoms, a relationship influenced by the TIV.	[17]
Zhu et al., 2022	Cross-sectional	MDD patients (122) and healthy controls (119)	21–62	HAMD; HAMA	The results provide evidence supporting a potential link between vitamin D deficiency, alterations in functional brain network connectivity, and clinical symptoms in individuals diagnosed with MDD.	[61]

25(OH)D = 25-Hydroxyvitamin D; BDI = Beck Depression Inventory; BDI-I = Beck Depression Inventory I; BDI-II = Beck Depression Inventory II; CES-D = Center for Epidemiologic Studies-Depression; CLD = chronic liver diseases; DASS21 = Depression Anxiety Stress Scale; DSM-III = Diagnostic and Statistical Manual of Mental Disorders, third edition; DSM-IV = Diagnostic and Statistical Manual of Mental Disorders, fourth edition; DSM-V = Diagnostic and Statistical Manual of Mental Disorders, fifth edition; GDS = Geriatric Depression Scale; HAMA = 14-item Hamilton Rating Scale for Anxiety; HAMD = 24-item Hamilton Rating Scale for Depression; MDD = major depressive disorder; N = number; PHQ-9 = Patient Health Questionnaire; PTH = parathyroid hormone; Ref = reference; SCID-I = Structured Clinical Interview for DSM-IV Axis I Disorders; TIV = total intracranial volume.

**Table 2 pharmaceuticals-18-00792-t002:** Cross-sectional studies that have not highlighted the relationship between serum vitamin D levels and depression.

First Author, Year	Study Population (N)	Age (Years)	Depression Assessment Scale	Results	Ref.
Husemoen et al., 2016	Adult general Danish population (5308)	18–64	SCL-90-R	No association between 25(OH)D concentrations and symptoms/diagnosis of depression and anxiety was found.	[79]
Mousa et al., 2018	Overweight or obese and vitamin D-deficient adults (63)	18–60	BDI	The study did not show a relationship between serum 25(OH)D levels and depression.	[77]
Pan et al., 2009	Middle-aged and elderly Chinese individuals (3262)	50–70	CES-D	No association was found between serum vitamin D levels and depression in middle-aged and elderly Chinese individuals.	[76]
Yalamanchili et al., 2012	Older postmenopausal women (489)	65–77	GDS-Long Form 30	At baseline, depression was not associated with insufficient serum 25(OH)D levels.	[78]
Zhao et al., 2010	US adults (3916)	≥20	PHQ-9	The study indicated that no significant associations were found between serum concentrations of 25(OH)D and the presence of moderate-to-severe, major, or minor depression.	[75]

25(OH)D = 25-Hydroxyvitamin D; BDI = Beck Depression Inventory; CES-D = Center for Epidemiologic Studies-Depression; GDS = Geriatric Depression Scale; N = number; PHQ-9 = Patient Health Questionnaire; Ref = reference; SCL-90-R = Symptom Check List; US = United States.

**Table 3 pharmaceuticals-18-00792-t003:** Cohort studies that have highlighted the relationship between serum vitamin D levels and depression.

First Author, Year	Design of Study	Study Population (N)	Age (Years)	Follow-up (Years)	Depression Assessment Scale	Results	Ref.
Briggs et al., 2019	Longitudinal study	Nondepressed community-dwelling older people (3565)	≥50	4	CES-D	This study highlighted that hypovitaminosis D was associated with a 75% increased risk of developing depression over a four-year period.	[81]
May et al., 2010	Cohort study	Patients with a CV diagnosis (7358)	≥50	1.07 ± 1.13 (maximum 6.64)	ICD-9	The study highlighted a negative relationship between serum 25(OH)D levels and the incidence of depression in a sample of subjects without a prior history of depression.	[41]
Milaneschi et al., 2010	Cohort study	Older adults (954)	≥65	6	CES-D	Participants who had low levels of 25(OH)D at the beginning of the study experienced a more significant rise in depressive symptoms over the subsequent six years.	[80]
Milaneschi et al., 2014	Cohort study	Participants from the NESDA with current (1102) or remitted (790) depressive disorder and healthy controls (494)	18–65	2	DSM-IV; IDS	A negative relationship was observed between the presence and severity of depressive symptoms and serum vitamin D levels.	[51]

25(OH)D = 25-Hydroxyvitamin D; CES-D = Center for Epidemiological Studies-Depression Scale; CV = cardiovascular; DSM-IV = Diagnostic and Statistical Manual of Mental Disorders, fourth edition; ICD-9 = International Classification of Diseases, Ninth Edition; IDS = Inventory of Depressive Symptoms; N = number; NESDA = Netherlands Study of Depression and Anxiety; Ref = reference.

**Table 4 pharmaceuticals-18-00792-t004:** Cohort studies that have not highlighted the relationship between serum vitamin D levels and depression.

First Author, Year	Design of Study	Study Population (N)	Age (Years)	Follow-Up (Years)	Depression Assessment Scale	Results	Ref.
Berg et al., 2021	Prospective cohort study	Older patients with depression (232)	60–93	2	DSM-IV; IDS-SR	Over a period of two years, no independent association between serum 25(OH)D levels and the course of depression was found.	[82]
Chan et al., 2011	Cohort study	Community-dwelling Chinese men (629)	>65	4	GDS	Serum 25(OH)D concentration had no association with the incidence of depression at 4 years.	[72]
Jovanova et. al., 2017	Cohort study	Older adults (3251)	≥55	10	CES-D	No association was found between low vitamin D levels and changes in depressive symptomatology or the incidence of depression.	[74]

25(OH)D = 25-Hydroxyvitamin D; CES-D = Center for Epidemiological Studies-Depression Scale; DSM-IV = Diagnostic and Statistical Manual of Mental Disorders, fourth edition; GDS = Geriatric Depression Scale; IDS-SR = Inventory of Depressive Symptoms; N = number; Ref = reference.

**Table 5 pharmaceuticals-18-00792-t005:** Studies that have demonstrated the effectiveness of vitamin D supplementation in relation to MDD.

First Author, Year (Ref.)	Study Design	Study Participants, Age (Years)	Sample Size		Intervention	Follow-up Duration	Depression Assessment Scale	Results
			Experiment	Placebo				
Abiri et al., 2022 [5]	Randomised, double-blind, placebo-controlled clinical trial	Obese women with mild-to-moderate depressive symptoms, 20–45	Group 1: 27 Group 2: 27 Group 3: 27	Group 4: 27	Group 1: 50,000 IU vitamin D soft gel/wk + 250 mg magnesium tablet/day Group 2: 50,000 IU vitamin D soft gel/wk + magnesium placebo/day Group 3: vitamin D placebo/wk + 250 mg magnesium tablet/day Group 4: vitamin D placebo/wk + magnesium placebo/day	8 wks	BDI-II	Positive effects on depressive symptoms were found in vitamin D plus magnesium supplementation.
Alavi et al., 2019 [88]	Randomised, placebo-controlled trial	Older adults with moderate-to-severe depression, ≥ 60	39	39	50,000 IU vitamin D_3_/wk + TAU or placebo + TAU	8 wks	GDS-15	The severity of depression symptoms improved after vitamin D supplementation.
Alghamdi et al., 2020 [49]	Randomised clinical trial	Male and female patients diagnosed with MDD, 18–65	49	13	50,000 IU vitamin D_3_/wk + SOC or SOC	3 months	BDI	Female patients showed a more significant improvement compared to male patients in their depressive symptoms.
Amini et al., 2022 [104]	Randomised, double-blind, placebo-controlled clinical trial	Women with PPD, 18–45	Group 1: 27 Group 2: 27	Group 3: 27	Group 1: 50,000 IU vitamin D_3_ fortnightly + 500 mg calcium carbonate daily Group 2: 50,000 IU vitamin D_3_ fortnightly + placebo of calcium daily Group 3: Placebo of vitamin D_3_ fortnightly + placebo of calcium daily	8 wks	EPDS	Vitamin D supplementation highlighted an increase in serum 25(OH)D concentration, which improved PPD symptoms.
Dabbaghmanesh et al., 2019 [100]	Randomised, placebo-controlled clinical trial	Nulliparous and multiparous females, ≥ 18	46	52	2000 IU vitamin D_3_ daily or placebo	26th to 28th week of gestation until birth	EPDS	Vitamin D supplementation in late pregnancy may result in a substantial change in postnatal depression at the fourth week.
Ghaderi et al., 2017 [97]	Randomised, double-blind, placebo-controlled clinical trial	MMT patients, 25–70	34	34	50,000 IU vitamin D/2 wks or placebo	12 wks	BDI; PSQI; BAI	Salutary effects on psychological symptoms were found in MMT patients after vitamin D supplementation.
Ghaderi et al., 2020 [98]	Randomised, double-blind, placebo-controlled clinical trial	MMT patients, 18–60	32	32	50,000 IU vitamin D/2 wks or placebo	24 wks	BDI; BAI	Improvements in cognitive function and some mental health parameters were observed after vitamin D supplementation in MMT patients.
Jamilian et al., 2018 [101]	Randomised, double-blind, placebo-controlled clinical trial	Women with polycystic ovary syndrome, 18–40	30	30	50,000 IU vitamin D/2 wks + 2000 mg/day omega-3 fatty acid from fish oil or placebo	12 wks	BDI; DASS; GHQ-28	Mental health parameters improved following vitamin D plus omega-3 fatty acid supplementation.
Kaviani et al., 2020 [85]	Double-blind, randomised clinical trial	Patients with mild-to-moderate depression, 18–60	28	28	50,000 IU cholecalciferol/2 wks or placebo	8 wks	BDI-II	An increased concentration of 25(OH)D led to a decrease in depression severity in patients diagnosed with mild-to-moderate depression after vitamin D supplementation.
Kaviani et al., 2022 [12]	Double-blind, randomised clinical trial	Patients with mild-to-moderate depression, 18–60	28	28	50,000 IU cholecalciferol/2 wks or placebo	8 wks	BDI-II	Individuals diagnosed with mild-to-moderate depression exhibited a substantial decrease in BDI-II scores after vitamin D supplementation.
Khoraminya et al., 2013 [89]	Double-blind, randomised, placebo-controlled trial	Patients with a diagnosis of MDD, 18–65	20	20	Daily 1500 IU vitamin D_3_ + 20 mg fluoxetine or vitamin D placebo + 20 mg fluoxetine	8 wks	HDRS; BDI	Co-administration of vitamin D and fluoxetine demonstrated superior effects compared to fluoxetine alone from the fourth week of treatment.
Lv et al., 2024 [99]	Retrospective, observational study	Pregnant women diagnosed with vitamin D deficiency at 12–14 weeks of gestation; 28–34	1365	-	800 IU daily from 14 weeks onwards	14 wks onwards until gestational week 39 (38, 39) prior to delivery	HDRS; SDS; EPDS	Pregnant women in the 12–14 weeks gestational period with vitamin D deficiency had insufficient levels of vitamin D, and showed an improvement in depressive symptoms following supplementation.
Mozaffari-Khosravi et al., 2013 [86]	Randomised clinical trial	Depressed patients with vitamin D deficiency; 20–60	Group 1: 40 Group 2: 40	Group 3: 40	Group 1: 300,000 IU of vitamin D intramuscularly Group 2: 150,000 IU of vitamin D intramuscularly Group 3: received nothing	3 months	BDI-II	Vitamin D deficiency is highly prevalent among patients with depression, and restoring its levels leads to an improvement in depressive symptoms. Higher doses, such as 300,000 IU compared to 150,000 IU, are more efficient and also safe.
Narula et al., 2017 [93]	Double-blind, randomised, placebo-controlled trial	Patients with a prior diagnosis of CD in remission, 18–70	Group 1: 18 Group 2: 16	-	Group 1: 10,000 IU vitamin D_3_/day Group 2: 1000 IU vitamin D_3_/day	12 months	HADS	Both doses indicated mood improvement, with the higher dose exhibiting potential in normalising mood symptoms in patients suffering from clinical anxiety and depression.
Omidian et al., 2019 [90]	Double-blind, randomised, placebo-controlled trial	T2DM patients with mild-to-moderate depression, 30–60	32	34	4000 IU vitamin D/day or placebo	12 wks	Persian version of BDI-II	Patients with T2DM and depression showed reduced depressive symptoms after vitamin D supplementation.
Penckofer et al., 2017 [91]	Open-label, proof-of-concept study	Women with T2DM and significant depressive symptoms, ≥ 18	50	-	50,000 IU ergocalciferol/wk	6 months	CES-D; PHQ-9; STAI; SF-12	Reduced depressive symptoms were emphasised following vitamin D_2_ supplementation in women with T2DM.
Raygan et al., 2018 [103]	Double-blind, randomised, placebo-controlled trial	Diabetic people with CHD, 45–85	30	30	50,000 IU vitamin D_3_ every 2 wks plus 8 × 10^9^ CFU/g probiotic or placebo	12 wks	BDI; BAI; GHQ-28	Mental health parameters were considerably enhanced in diabetic patients with CHD following co-supplementation with vitamin D and probiotics.
Sharifi et al., 2019 [94]	Double-blind, randomised, placebo-controlled trial	Patients with mild-to-moderate UC, vitamin D group: 37.5 ± 9.0; placebo group: 35.0 ± 9.2	46	40	One muscular injection of 1 mL 300,000 IU vitamin D_3_ or 1 mL normal saline as placebo	3 months	BDI-II	Mild-to-moderate UC patients receiving 300,000 IU vitamin D_3_ showed a major reduction in BDI-II scores after 3 months.
Sikaroudi et al., 2020 [35]	Double-blind, randomised, placebo-controlled trial	Male and female patients with IBS-D and insufficient vitamin D levels, 18–65	39	35	50,000 IU vitamin D_3_/wk or placebo	9 wks	HADS	Considerable differences were observed between the groups, with patients receiving vitamin D showing a decrease in depressive symptoms.
Stokes et al., 2016 [68]	Interventional study	Patients with CLD and inadequate vitamin D concentrations, 20–81	77	-	20,000 IU vitamin D_3_ daily for the first seven days, then weekly thereon	6 months	BDI-II	Normalisation of vitamin D levels improved depressive symptoms in patients diagnosed with CLD.
Vaziri et al., 2016 [16]	Randomised clinical trial	Pregnant women with a baseline depression score of 0 to 13, 18–39	78	75	2000 IU vitamin D_3_/day or placebo	From 26 to 28 wks of gestation until childbirth	EPDS	Daily vitamin D_3_ supplementation, from 26 to 28 weeks of gestation until childbirth, indicated a decrease in perinatal depression levels.
Vellekkatt et al., 2020 [87]	Double-blind, randomised, parallel-arm, placebo-controlled trial	Patients with MDD and concurrent vitamin D deficiency, 18–65	23	23	TAU + single parenteral dose of 300,000 IU of cholecalciferol or TAU + placebo	12 wks	HDRS-17; QLES; CGI-S	A higher, single-administration dose of 300,000 IU of vitamin D was proven to be efficient in treating MDD and showed improvement in short-term quality of life.
Yosaee et al., 2020 [102]	Double-blind, randomised, placebo-controlled trial	Obese/overweight patients with depressive symptoms, > 20	Group 1: 27 Group 2: 24 Group 3: 25	Group 4: 22	Group 1: 2000 IU/day vitamin D_3_ + daily placebo for zinc Group 2: 30 mg/day zinc gluconate + daily placebo for vitamin D Group 3: 2000 IU/day vitamin D_3_ + 30 mg/day zinc gluconate Group 4: vitamin D placebo + zinc placebo	12 wks	BDI-II	Obese and overweight patients with depressive symptoms showed a significant decrease in BDI-II scores after administration of vitamin D, zinc, and both concurrently.
Zaromytidou et al., 2022 [96]	Randomised controlled study	Elderly people with prediabetes, > 60	45	45	25,000 IU vitamin D_3_/wk or nothing	12 months	PHQ-9: STAI	Relieved symptoms of anxiety and depression were observed in a high-risk population after vitamin D dosing.
Zheng et al., 2019 [95]	Double-blind, randomised, parallel-arm, placebo-controlled trial	Patients with knee OA and vitamin D deficiency, vitamin D group: 63.5; placebo group: 62.9	209	204	50,000 IU vitamin D_3_/month or placebo	24 months	PHQ-9	Depressive patients diagnosed with knee OA whose vitamin D levels were sustained through supplementation demonstrated positive effects on depressive symptoms.
Zhou et al., 2024 [92]	Prospective study	Patients with DPN and vitamin D insufficiency, ≥ 60	158	-	5000 IU vitamin D daily	12 wks	HAMD-17	The study results highlighted the beneficial effects of vitamin D supplementation in patients with DPN and vitamin D insufficiency.

25(OH)D = 25-Hydroxyvitamin D; BAI = Beck Anxiety Inventory; BDI = Beck Depression Inventory; BDI-II = Beck Depression Inventory-II; CD = Crohn’s disease; CES-D = Center for Epidemiologic Studies Depression; CFU = colony-forming units; CGI-S = Clinical Global Impression severity of illness; CHD = coronary heart disease; CLD = chronic kidney disease; DASS = Depression Anxiety Stress Scale; DPN = diabetic peripheral neuropathy; EPDS = Edinburgh Postnatal Depression scale; GDS-15 = Geriatric Depression Scale-15; GHQ-28 = General health questionnaire-28; HADS = Hospital Anxiety and Depression Scale; HAMD-17 = 17-item Hamilton Rating Scale for Depression; HDRS = Hamilton Depression Rating Scale; IBS-D = irritable bowel syndrome—diarrhoea-predominant; MDD = major depressive disorder; MMT = maintenance methadone treatment; OA = osteoarthritis; PHQ-9 = Patient Health Questionnaire; PPD = postpartum depression; PSQI = Pittsburgh Sleep Quality Index; QLES = Quality of Life Enjoyment and Satisfaction Questionnaire-Short Form; Ref = reference; SDS = Zung Self-Rating Depression Scale; SF-12 = Short Form (mental and physical health status); SOC = standard of care (including pharmacological treatment and psychological support); STAI = State-Trait Anxiety Inventory; T2DM = type 2 diabetes mellitus; TAU = treatment as usual; wk/s = week/s; UC = ulcerative colitis.

**Table 6 pharmaceuticals-18-00792-t006:** Studies that have not demonstrated the effectiveness of vitamin D supplementation in relation to MDD.

First Author, Year (Ref.)	Study Design	Study Participants, Age (Years)	Sample Size		Intervention	Follow-up Duration	Depression Assessment scale	Results
			Experiment	Placebo				
Choukri et al., 2018 [109]	Randomised, double-blind, placebo-controlled clinical trial	Healthy women, 18–40	76	76	50,000 IU vitamin D_3_/month or placebo	6 months	CES-D; HADS; Flourishing Scale	Depression and other mood outcomes in healthy premenopausal women were not positively influenced by vitamin D supplementation.
Hansen et al., 2019 [11]	Randomised, double-blind, placebo-controlled trial	Patients diagnosed with mild-to-severe depression, 18–65	28	34	70 µg (2800 IU) vitamin D_3_/day + TAU or placebo + TAU	6 months	HAMD-17	Symptom severity did not show any decline in patients with depression following vitamin D supplementation.
Jorde et al., 2018 [111]	Randomised, double-blind, placebo-controlled trial	Subjects recruited from the Tromsø Study (a population-based health survey in the municipality of Tromsø), ≥ 40	206	202	100,000 IU vitamin D as bolus dose, followed by 20,000 IU vitamin D/wk or placebo	4 months	BDI-II	The results of the study highlighted no effects of vitamin D supplementation on depressive symptoms.
Koning et al., 2019 [112]	Randomised, double-blind, placebo-controlled clinical trial	An older population with a low vitamin D status and clinically relevant depressive symptoms, 60–80	77	78	Daily dose of 1200 IU vitamin D_3_ or placebo	12 months	CES-D	No impact on depressive symptoms was observed after vitamin D supplementation.
Marsh et al., 2017 [22]	Randomised, double-blind, placebo-controlled trial	Patients diagnosed with bipolar disorder (currently experiencing depressive symptoms) and vitamin D deficiency, 18–70	16	17	5000 IU vitamin D_3_/day or placebo	12 wks	MADRS; HAM-A; YMRS	High levels of 25(OH)D, compared to a placebo, did not lead to any improvement in depressive symptoms.
Mason et al., 2016 [110]	Randomised, double-blind, placebo-controlled clinical trial	Overweight postmenopausal women, 50–75	109	109	Weight loss + 2000 IU vitamin D_3_/day or weight loss + daily placebo	12 months	BSI; PQSI	Vitamin D supplementation, compared to a placebo, did not have an impact on depressive symptoms.
Mirzavandi et al., 2020 [116]	Randomised clinical trial	Patients with T2DM and vitamin D deficiency, 30–60	25	25	200,000 IU of vitamin D injection at week 0 and week 4 of study or nothing	8 wks	Beck depression test	Supplemented high levels of serum 25(OH)D had no effect on the Beck depression score.
Mousa et al., 2018 [77]	Randomised, double-blind, placebo-controlled clinical trial	Overweight or obese and vitamin D-deficient adults, 18–60	26	22	Bolus oral dose of 100,000 IU followed by 4000 IU daily of cholecalciferol or placebo	16 wks	BDI	No noticeable differences were observed in the total BDI scores between patients supplemented with vitamin D and those in the placebo group.
Okereke et al., 2020 [117]	Double-blind, placebo-controlled randomised trial	Men and women in the VITAL-DEP, ≥ 50	9181	9172	2000 IU of cholecalciferol/day and fish oil (Omacor; 1 g/day capsule containing 840 mg of omega-3 fatty acids as 465 mg of eicosapentaenoic acid and 375 mg of docosahexaenoic acid) or placebo	5.3 years	PHQ-8	Patients who received vitamin D supplementation compared to a placebo exhibited no major differences in the incidence and recurrence of depression.
Rahman et al., 2023 [107]	Randomised, double-blind, placebo-controlled clinical trial	Older Australian adults, 60–84	10,662	10,653	60,000 IU vitamin D_3_/month or placebo	5 years	PHQ-9	This analysis did not find an overall improvement in depressive symptoms following monthly vitamin D supplementation.
Rolf et al., 2017 [114]	Randomised, placebo-controlled trial	MS patients, 18–55	20	20	7000 IU cholecalciferol/day in first 4 weeks, followed by 14,000 IU cholecalciferol/day up to week 48 or placebo	48 wks	HADS-D; FSS	High doses of vitamin D supplementation show no decline in depressive symptoms.
Wang et al., 2016 [113]	Prospective, randomised, double-blind trial	Dialysis patients with depressive symptoms and vitamin D insufficiency, ≥ 18	362	364	50,000 IU vitamin D_3_/wk or placebo	52 wks	Chinese version of BDI-II	No beneficial effect on depressive symptoms was observed after vitamin D supplementation in dialysis patients.
Yalamanchili et al., 2012 [78]	Double-blind, randomised, placebo-controlled clinical trial	Older postmenopausal women, 65–77	Group 1: 120 Group 2: 123 Group 3: 122	Group 4: 123	Group 1: HT (oestrogens 0.625 mg/daily in hysterectomized women or combined with medroxyprogesterone acetate 2.5 mg/daily in women with intact uterus) Group 2: calcitriol 0.25 g BID Group 3: HT + calcitriol Group 4: matching placebo	3 years	GDS-Long form 30	No significant effect was observed after HT, calcitriol, or their co-administration in older postmenopausal women.
Yalamanchili et al., 2018 [108]	Double-blind, randomised, placebo-controlled clinical trial	Older Caucasian and African American women from Gallagher et al. [45,50], 57–90	Group 1: 22 (20 Caucasian, 2 Afro-American) Group 2: 45 (21 Caucasian, 24 Afro-American) Group 3: 43 (20 Caucasian, 23 Afro-American) Group 4: 44 (21 Caucasian, 23 Afro-American) Group 5: 23 (20 Caucasian, 3 Afro-American) Group 6: 24 (20 Caucasian, 4 Afro-American) Group 7: 34 (20 Caucasian, 14 Afro-American)	Group 8: 38 (21 Caucasian, 17 Afro-American)	Group 1: 400 IU of vitamin D_3_/day Group 2: 800 IU of vitamin D_3_/day Group 3: 1600 IU of vitamin D_3_/day Group 4: 2400 IU of vitamin D_3_/day Group 5: 3200 IU of vitamin D_3_/day Group 6: 4000 IU of vitamin D_3_/day Group 7: 4800 IU of vitamin D_3_/day Group 8: placebo	12 months	GDS	Depression scores were not positively influenced in older Caucasian and African American women following vitamin D supplementation.
Zhang et al., 2018 [115]	Randomised, double-blind clinical trial	PTB patients with depression, ≥ 18	56	64	Bolus oral dose of 100,000 IU cholecalciferol/wk or placebo	8 wks	BDI-II	No significant effect on BDI-II scores was highlighted following vitamin D supplementation in PTB patients with depression.

25(OH)D = 25-Hydroxyvitamin D; BDI = Beck Depression Inventory; BDI-II = Beck Depression Inventory-II; BID = bis in die (twice a day); BSI = Brief Symptom Inventory-18; CES-D = Center for Epidemiologic Studies Depression Scale; FSS = Fatigue Severity Scale; GDS = Geriatric depression score; HADS = Hospital Anxiety and Depression Scale; HADS-D = Hospital Anxiety and Depression Scale (HADS) depression scale (HADS-D); HAM-A = Hamilton Anxiety Rating Scale; HAMD-17 = 17-item Hamilton Rating Scale for Depression; HT = hormone therapy; MADRS = Montgomery–Åsberg Depression Rating Scale; MS = multiple sclerosis; PHQ-8/9 = Patient Health Questionnaire; PQSI = Pittsburg Sleep Quality Index; PTB = pulmonary tuberculosis; Ref = reference; T2DM = type 2 diabetes mellitus; TAU = treatment as usual; VITAL-DEP = Vitamin D and Omega-3 Trial-Depression Endpoint Prevention; wk/s = week/s; YMRS = Young Mania Rating Scale.

## Data Availability

No new data were created or analyzed in this study.

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
