# Peer review of "The Role of Vitamin D in the Management of Major Depressive Disorder: A Systematic Review"

_pharmaceuticals, 2025, doi:10.3390/ph18060792_

Round 1
Reviewer 1 Report
Comments and Suggestions for Authors
The main reasons for an increasing interest to develop anti-depressive drugs is
the low response rate to pharmacological treatment, the significant adverse effects of medication, and the debilitating impact of depressive disorder on mental health.
Since there is some evidence that Vitamin D has a neuroimmunomodulatory effect, the authors have screened the existing literature for its use in depression, which may represent a new strategy for the prevention and/or treatment of depressive disorder. Specifically, the authors reviewed the relationship between serum 25(OH)D levels and depression to evaluate the effect of vitamin D supplementation on depressive symptoms.
Some but no conclusive evidence from the scientific literature suggests the beneficial potential of vitamin D supplementation both as a prophylactic strategy to reduce the risk of developing depression and as an adjunctive therapy for symptom control.
A typical conclusion that there is no clear conclusion is the final statement: “This systematic review also aims to raise awareness about the impact of vitamin D on mental health and highlight the importance of the topic for future research”.
However, due to the many publications with contradicting results presented, the awareness is there, but the situation remains inconclusive. Since this confirms the present state of knowledge son that the information is provided is of no added value, and I do not recommend publication.
Author Response
Esteemed reviewer,
Thank you very much for your detailed and thoughtful feedback. The authors sincerely appreciate the time, effort and consideration dedicated to evaluating the manuscript entitled THE ROLE OF VITAMIN D IN MANAGEMENT OF MAJOR DEPRESSIVE DISORDER: A SYSTEMATIC REVIEW.
Comment: [The main reasons for an increasing interest to develop anti-depressive drugs is the low response rate to pharmacological treatment, the significant adverse effects of medication, and the debilitating impact of depressive disorder on mental health.
Since there is some evidence that Vitamin D has a neuroimmunomodulatory effect, the authors have screened the existing literature for its use in depression, which may represent a new strategy for the prevention and/or treatment of depressive disorder. Specifically, the authors reviewed the relationship between serum 25(OH)D levels and depression to evaluate the effect of vitamin D supplementation on depressive symptoms.
Some but no conclusive evidence from the scientific literature suggests the beneficial potential of vitamin D supplementation both as a prophylactic strategy to reduce the risk of developing depression and as an adjunctive therapy for symptom control.
A typical conclusion that there is no clear conclusion is the final statement: “This systematic review also aims to raise awareness about the impact of vitamin D on mental health and highlight the importance of the topic for future research”.
However, due to the many publications with contradicting results presented, the awareness is there, but the situation remains inconclusive. Since this confirms the present state of knowledge son that the information is provided is of no added value, and I do not recommend publication.]
Response: [The authors fully acknowledge the concern regarding the inconclusive nature of the current evidence on the role of vitamin D in depression. Indeed, part of the challenge in addressing this topic is the complexity of existing data, which includes many studies with varying designs, populations, and outcomes. However, it is precisely this fragmented and often contradictory landscape that underscores the need for a comprehensive and up-to-date synthesis of the available evidence.
The purpose of this systematic review was not only to explore the potential role of vitamin D in depression, but also to map out the current state of the literature and identify where gaps, inconsistencies, and methodological limitations persist. While the findings do not allow for a definitive conclusion, the review provides a structured, transparent, and extensive overview of the available studies—including cross-sectional, cohort, and randomised controlled trials—up to and including 2024.
Additionally, this review serves as a bibliometric analysis that lays the foundation for a new practical study currently underway, involving patient data. This marks the beginning of a more applied research phase aimed at generating original clinical findings.
Moreover, this review adds value by:
Including both observational and interventional data, this review is among the few that integrate evidence from cross-sectional, cohort, and randomized controlled trials, thereby offering a broader and more comprehensive perspective
Providing a detailed comparative analysis of study characteristics, population diversity, and assessment tools used, this review highlights the heterogeneity across studies, which helps readers interpret why the literature remains inconclusive.
Identifying methodological patterns and limitations that may explain the contradictions in the field, which is essential for guiding future research.
The authors believe that acknowledging the lack of a clear conclusion is not a weakness, but a reflection of scientific integrity. It highlights the importance of cautious interpretation, the need for well-designed future studies, and the relevance of the topic in current mental health research—especially given the global burden of depression and the limitations of current treatment options.
The authors have revised the manuscript to strengthen these aspects and have clarified the contributions of the review. They respectfully hope that, upon reconsideration, the manuscript may be seen as a useful and timely resource for researchers and clinicians navigating this complex and evolving topic.]
The authors sincerely thank you for your valuable suggestions and insightful guidance, which have been greatly appreciated.
I remain most respectfully yours,
Andreea Roșian et al
Reviewer 2 Report
Comments and Suggestions for Authors
This manuscript presents a systematic review focused on the relationship between vitamin D status and major depressive disorder (MDD), including an evaluation of the impact of vitamin D supplementation. The topic is both clinically significant and timely, given the increasing interest in non-traditional adjunctive therapies for depression. The authors have compiled a substantial body of literature, and the manuscript reflects considerable effort in gathering and synthesizing current evidence.
The title is appropriate and accurately reflects the content of the paper. The abstract outlines the rationale and conclusions of the review clearly, but it lacks structured subheadings typically expected in systematic reviews, such as Background, Methods, Results, and Conclusion. Additionally, it does not mention how many studies were included, the inclusion period, or methodological details regarding the study selection process.
The introduction is thorough and sets up the rationale for the review well. It covers both the pathophysiology of depression and the proposed roles of vitamin D, including biochemical pathways. However, it could benefit from condensation to avoid repetition, especially as some of the content reappears in later sections.
The methods section is the primary area of concern. While the authors list databases and keywords used, the review does not conform to established standards for systematic reviews, such as PRISMA. There is no PRISMA flow diagram, no mention of protocol registration (e.g., PROSPERO), no details on the number of independent reviewers involved in screening and data extraction, and no quality assessment of the included studies. The lack of a risk of bias assessment (such as the Newcastle-Ottawa Scale or Cochrane Risk of Bias Tool) limits the ability to interpret the strength of the evidence. Without these components, the review is better classified as a narrative review with a structured search strategy rather than a systematic review.
The results section is very detailed and includes a large number of studies, which are categorized into cross-sectional, cohort, and interventional designs. Tables are used effectively to present the data, and the inclusion of both positive and null findings is commendable. However, the narrative accompanying the tables is often repetitive and overly lengthy, making it difficult to extract key findings efficiently. Structuring the results more tightly around clear thematic subheadings and reducing redundant text would improve clarity.
The discussion summarizes the key findings and mechanisms well, including plausible biological pathways through which vitamin D might influence depressive symptoms. However, the authors lean heavily toward interpreting the evidence as supportive of vitamin D’s antidepressant effects, despite acknowledging that many studies report null or contradictory findings. This introduces a potential bias in the interpretation. The review would benefit from a more balanced appraisal, particularly regarding the limited number and inconsistent quality of randomized controlled trials.
The conclusion is cautiously optimistic but could be reworded to reflect the methodological limitations and heterogeneity of existing evidence more explicitly. Phrases such as “vitamin D could represent a new strategy for prevention and/or treatment” should be tempered to reflect the current level of evidence, which is suggestive but not definitive.
The references are broad and up-to-date, and the tables are informative and well-organized. However, formatting inconsistencies should be addressed, including missing or incomplete DOIs.
In terms of strengths, the manuscript covers a clinically relevant topic, incorporates a wide range of studies, and presents a compelling mechanistic rationale. The scope is comprehensive, and including observational and interventional evidence is a significant asset. The detailed tables are handy for readers seeking to compare studies.
Regarding significant weaknesses, the lack of adherence to systematic review methodology standards is the most serious. Without a risk of bias assessment or protocol registration, the credibility of the findings is reduced. The narrative style is also too lengthy and sometimes redundant, affecting readability and undermining the message's clarity.
In summary, this manuscript addresses a topic of considerable clinical interest and compiles a wealth of literature on the subject. However, a major revision is required to meet the standards of a publishable systematic review. The authors must align the methodology with PRISMA guidelines, provide a flowchart of study selection, include a risk of bias assessment, and present a more balanced and concise narrative.
Author Response
Esteemed reviewer,
Thank you very much for your detailed and thoughtful feedback. The authors sincerely appreciate the time, effort and consideration dedicated to evaluating the manuscript entitled THE ROLE OF VITAMIN D IN MANAGEMENT OF MAJOR DEPRESSIVE DISORDER: A SYSTEMATIC REVIEW.
Comment 1: [The title is appropriate and accurately reflects the content of the paper. The abstract outlines the rationale and conclusions of the review clearly, but it lacks structured subheadings typically expected in systematic reviews, such as Background, Methods, Results, and Conclusion. Additionally, it does not mention how many studies were included, the inclusion period, or methodological details regarding the study selection process.]
Response: [Thank you very much. We appreciate the time and effort that you have dedicated to providing your valuable feedback on our manuscript. In the revised version of the abstract, we have incorporated structured subheadings and provided additional details, including the number of studies reviewed, the time frame of inclusion, and a brief description of the selection methodology to enhance transparency and alignment with systematic review conventions.]
Comment 2: [The introduction is thorough and sets up the rationale for the review well. It covers both the pathophysiology of depression and the proposed roles of vitamin D, including biochemical pathways. However, it could benefit from condensation to avoid repetition, especially as some of the content reappears in later sections.]
Response: [Thank you for this comment. In the revised version, the introduction has been carefully reviewed and updated to improve conciseness and eliminate repetition. Overlapping content that appeared again in later sections has been reduced or rephrased to ensure a clearer and more streamlined presentation of the rationale for the review.]
Comment 3: [The methods section is the primary area of concern. While the authors list databases and keywords used, the review does not conform to established standards for systematic reviews, such as PRISMA. There is no PRISMA flow diagram, no mention of protocol registration (e.g., PROSPERO), no details on the number of independent reviewers involved in screening and data extraction, and no quality assessment of the included studies. The lack of a risk of bias assessment (such as the Newcastle-Ottawa Scale or Cochrane Risk of Bias Tool) limits the ability to interpret the strength of the evidence. Without these components, the review is better classified as a narrative review with a structured search strategy rather than a systematic review.]
Response: [Thank you for this comment. The authors appreciate the emphasis placed on adhering to standardized guidelines for systematic reviews.
In response, the revised version of the manuscript now includes a PRISMA flow diagram to provide a clear overview of the study selection process. Additionally, the number of independent reviewers involved in screening and data extraction has been specified; in this case, a single reviewer conducted the process, and this has been clearly stated to ensure transparency.
The authors fully acknowledge that the review does not include a registered protocol (e.g., in PROSPERO), as registration is no longer accepted after data extraction has begun. This limitation is recognized and will be taken into account for future research efforts.
Similarly, we acknowledge the absence of a formal risk of bias assessment in the current version. Given the advanced stage of the review at the time of this realization, the authors plan to conduct a risk of bias evaluation in a timely manner, and this can be incorporated into the manuscript if additional time is granted.
The authors are committed to maintaining the highest standards of scientific rigor and have made revisions accordingly to better align the review with systematic review methodology.]
Comment 4: [The results section is very detailed and includes a large number of studies, which are categorized into cross-sectional, cohort, and interventional designs. Tables are used effectively to present the data, and the inclusion of both positive and null findings is commendable. However, the narrative accompanying the tables is often repetitive and overly lengthy, making it difficult to extract key findings efficiently. Structuring the results more tightly around clear thematic subheadings and reducing redundant text would improve clarity.]
Response: [Thank you for this comment. In the revised version, the narrative accompanying the tables has been carefully reviewed and modified. Redundant content has been reduced, and the descriptions of individual studies have been condensed to avoid repetition. These adjustments were made to improve clarity and allow for a more concise and focused presentation of the findings, while maintaining the necessary level of detail.]
Comment 5: [The discussion summarizes the key findings and mechanisms well, including plausible biological pathways through which vitamin D might influence depressive symptoms. However, the authors lean heavily toward interpreting the evidence as supportive of vitamin D’s antidepressant effects, despite acknowledging that many studies report null or contradictory findings. This introduces a potential bias in the interpretation. The review would benefit from a more balanced appraisal, particularly regarding the limited number and inconsistent quality of randomized controlled trials.]
Response: [Thank you for this comment. The discussion section has been revised to ensure a more balanced and objective interpretation of the findings. Sentences that previously emphasized the positive effect of vitamin D have been removed or rephrased to better reflect the variability and limitations of the existing evidence. The revised version maintains a neutral tone, acknowledging both the potential and the uncertainties surrounding the role of vitamin D in depression, with particular attention to the limited number and inconsistent quality of randomized controlled trials.]
Comment 6: [The conclusion is cautiously optimistic but could be reworded to reflect the methodological limitations and heterogeneity of existing evidence more explicitly. Phrases such as “vitamin D could represent a new strategy for prevention and/or treatment” should be tempered to reflect the current level of evidence, which is suggestive but not definitive.]
Response: [Thank you for this comment. The authors have revised the conclusion to adopt a more balanced tone, ensuring that the statements reflect the current state of the evidence. Optimistic language has been moderated to acknowledge the limitations in study design, variability among results, and the need for further high-quality research before drawing firm clinical implications.]
Comment 7: [The references are broad and up-to-date, and the tables are informative and well-organized. However, formatting inconsistencies should be addressed, including missing or incomplete DOIs]
Response: [Thank you for this comment. Indeed, two references are missing DOIs. The first reference is a book, which does not have a DOI, but if needed, we can add the ISBN number. The second reference is a web link (https://ods.od.nih.gov/factsheets/VitaminD-HealthProfessional/).]
Comment 8: [Regarding significant weaknesses, the lack of adherence to systematic review methodology standards is the most serious. Without a risk of bias assessment or protocol registration, the credibility of the findings is reduced. The narrative style is also too lengthy and sometimes redundant, affecting readability and undermining the message's clarity.]
Response: [Thank you for this comment. The authors have revised and condensed the narrative to improve clarity and eliminate redundancy. As previously mentioned, the revised manuscript now includes a PRISMA flow diagram and specifies that screening and data extraction were conducted by a single reviewer. While protocol registration was not possible due to the review’s completion stage, this limitation is acknowledged. Similarly, although a formal risk of bias assessment was not initially included, the authors intend to conduct and incorporate it if time allows. The manuscript has been updated to more closely reflect systematic review standards.]
Comment 9: In summary, this manuscript addresses a topic of considerable clinical interest and compiles a wealth of literature on the subject. However, a major revision is required to meet the standards of a publishable systematic review. The authors must align the methodology with PRISMA guidelines, provide a flowchart of study selection, include a risk of bias assessment, and present a more balanced and concise narrative.]
Response: [Thank you for this comment. We fully acknowledge the importance of adhering to systematic review standards. In the revised version of the manuscript, the methodology has been aligned with PRISMA guidelines, a detailed flowchart of the study selection process has been included, and the narrative has been revised to ensure greater balance and conciseness. As previously mentioned, the only remaining component not yet included is the formal risk of bias assessment, which the authors are prepared to conduct and incorporate if additional time is allowed.]
The authors sincerely thank you for your valuable suggestions and insightful guidance, which have been greatly appreciated.
I remain most respectfully yours,
Andreea Roșian et al
Reviewer 3 Report
Comments and Suggestions for Authors
This article addresses a very important topic concerning relation of vit D and depression. Writing is very clear and organized. The review concludes that higher 25(OH)D levels are inversely related to depression risk and that supplementation may improve symptoms. Great efforts were done. However, many revisions are required.
- For the abstract , in methods subsection [In this regard, relevant articles were searched on platforms such as PubMed, MDPI, ResearchGate, Springer Link, Springer Open, and ScienceDirect. ] add time frame .
- Researchgate database inclusion in this study should be explained as it is not reliable one.
- In the abstract the authors should highlight novelty over related review articles such as
Srifuengfung, M., Srifuengfung, S., Pummangura, C., Pattanaseri, K., Oon-Arom, A. and Srisurapanont, M., 2023. Efficacy and acceptability of vitamin D supplements for depressed patients: A systematic review and meta-analysis of randomized controlled trials. Nutrition, 108, p.111968.
Mikola, T., Marx, W., Lane, M.M., Hockey, M., Loughman, A., Rajapolvi, S., Rocks, T., O’Neil, A., Mischoulon, D., Valkonen-Korhonen, M. and Lehto, S.M., 2023. The effect of vitamin D supplementation on depressive symptoms in adults: A systematic review and meta‐analysis of randomized controlled trials. Critical reviews in food science and nutrition, 63(33), pp.11784-11801.
- In the abstract, it is much recommended to refer to which pharmaceutical formulation is recommended to achieve higher therapeutic activity in terms of lowering depression.
- At the beginning of section 3, it is strongly recommended to state how this section was divided, what are the main important subtitles and tables about. One or two precise paragraphs are enough.
- At the end of the discussion, the authors can add summary of key finding in each subsection and each table g. 3.1. The implication of vitamin D in pathophysiology of depression key points [ 3 or 4 sentences , highlights ]
- Provide precise future plan
- In conclusion, please provide a clear recommendation on optimal dosing, duration, or target populations for supplementation.
Author Response
Esteemed reviewer,
Thank you very much for your detailed and thoughtful feedback. The authors sincerely appreciate the time, effort and consideration dedicated to evaluating the manuscript entitled THE ROLE OF VITAMIN D IN MANAGEMENT OF MAJOR DEPRESSIVE DISORDER: A SYSTEMATIC REVIEW.
Comment 1: [For the abstract , in methods subsection [In this regard, relevant articles were searched on platforms such as PubMed, MDPI, ResearchGate, Springer Link, Springer Open, and ScienceDirect. ] add time frame.]
Response: [Thank you very much. We appreciate the time and effort that you have dedicated to providing your valuable feedback on our manuscript. In the revised version, the time frame of the literature search has been included in the abstract and also clearly mentioned in the Search Strategy section of the Methods chapter for clarity and completeness.]
Comment 2: [Researchgate database inclusion in this study should be explained as it is not reliable one.]
Response: [Thank you for this comment. Out of the 70 articles included in our review, only one was sourced from ResearchGate:
[100] M. H. Dabbaghmanesh, F. Vaziri, F. Najib, S. Nasiri, and S. Pourahmad, “The effect of vitamin D consumption during pregnancy on maternal thyroid function and depression: A randomized, placebo-controlled, clinical trial,” Jundishapur J Nat Pharm Prod, vol. 14, no. 2, May 2019, doi: 10.5812/JJNPP.65328.
Although ResearchGate is not a conventional primary database, this study was retrieved during the search and evaluated through the same selection process applied to all other articles. It was included following full-text screening, as it aligned with the objectives of the review and met the eligibility criteria.]
Comment 3: [In the abstract the authors should highlight novelty over related review articles such as
Srifuengfung, M., Srifuengfung, S., Pummangura, C., Pattanaseri, K., Oon-Arom, A. and Srisurapanont, M., 2023. Efficacy and acceptability of vitamin D supplements for depressed patients: A systematic review and meta-analysis of randomized controlled trials. Nutrition, 108, p.111968.
Mikola, T., Marx, W., Lane, M.M., Hockey, M., Loughman, A., Rajapolvi, S., Rocks, T., O’Neil, A., Mischoulon, D., Valkonen-Korhonen, M. and Lehto, S.M., 2023. The effect of vitamin D supplementation on depressive symptoms in adults: A systematic review and meta‐analysis of randomized controlled trials. Critical reviews in food science and nutrition, 63(33), pp.11784-11801.]
Response: [Thank you for this comment. The articles mentioned are indeed of high relevance, and we acknowledge their valuable contributions to the field. In fact, the review by Mikola et al. (2023) is included in our bibliography, and we have drawn upon some of the insights presented in their work.
One of the key distinctions of our review is its broader scope: while the cited articles focus exclusively on randomized controlled trials assessing the effect of vitamin D supplementation, our review also includes cross-sectional and cohort studies. This allows us to examine the relationship between vitamin D and depression beyond interventional studies. Additionally, our review incorporates studies published in both 2023 and 2024, ensuring the most recent literature is represented. This broader scope is now briefly highlighted in the abstract of the revised version.]
Comment 4: [[In the abstract, it is much recommended to refer to which pharmaceutical formulation is recommended to achieve higher therapeutic activity in terms of lowering depression.]
Response: [Thank you for this comment. While certain articles, such as the one by Srifuengfung et al. (2023), as mentioned in the previous comment, do specify which pharmaceutical formulation may be most effective, this was not the primary focus of our review. Our objective was to explore the relationship between serum 25-hydroxyvitamin D levels and depression, as well as the overall effect of vitamin D supplementation on depressive symptoms, rather than to compare or recommend specific formulations.]
Comment 5: [At the beginning of section 3, it is strongly recommended to state how this section was divided, what are the main important subtitles and tables about. One or two precise paragraphs are enough.]
Response: [Thank you for this comment. In the revised version, the beginning of the section was modified accordingly to include a brief overview. The section is now clearly divided into three main subtitles, each outlining a distinct aspect.]
Comment 6: [At the end of the discussion, the authors can add summary of key finding in each subsection and each table g. 3.1. The implication of vitamin D in pathophysiology of depression key points [ 3 or 4 sentences , highlights.]
Response: [Thank you for this comment. In response, the authors have added concise summaries at the end of each table, highlighting the key findings relevant to each set of studies. These summaries aim to enhance clarity, help readers quickly grasp the main outcomes, and support a more structured interpretation of the data presented throughout the review.]
Comment 7: [Provide precise future plan.]
Response: [Thank you for this comment. This review serves as a bibliometric analysis that lays the foundation for a new practical study currently underway, involving patient data. This marks the beginning of a more applied research phase aimed at generating original clinical findings.]
Comment 8: [In conclusion, please provide a clear recommendation on optimal dosing, duration, or target populations for supplementation.]
Response: [Thank you for this comment. In the revised version, the authors have included a more explicit summary in the conclusion regarding commonly reported dosages, durations, and target populations for vitamin D supplementation. While taking into account the limitations and variability of the existing studies, this addition aims to provide practical context while maintaining a cautious and evidence-based interpretation.]
The authors sincerely thank you for your valuable suggestions and insightful guidance, which have been greatly appreciated.
I remain most respectfully yours,
Andreea Roșian et al
Round 2
Reviewer 1 Report
Comments and Suggestions for Authors
No further comments.
Reviewer 3 Report
Comments and Suggestions for Authors
The authors did all required recommendations. I appreciate their efforts. the paper could be published in the current form.